# λ Recombineering Used to Engineer the Genome of Phage T7

**DOI:** 10.3390/antibiotics9110805

**Published:** 2020-11-13

**Authors:** Jordan D. Jensen, Adam R. Parks, Sankar Adhya, Alison J. Rattray, Donald L. Court

**Affiliations:** 1RNA Biology Laboratory, Center for Cancer Research, The National Cancer Institute at Frederick, Frederick, MD 21702, USA; jjensen5@tulane.edu (J.D.J.); adam.r.parks@gmail.com (A.R.P.); courtd@mail.nih.gov (D.L.C.); 2Laboratory of Molecular Biology, Center for Cancer Research, The National Cancer Institute, Bethesda, MD 20814, USA; adhyas@mail.nih.gov

**Keywords:** bacteriophage engineering, phage therapy, bacteriophage genetics, recombineering

## Abstract

Bacteriophage T7 and T7-like bacteriophages are valuable genetic models for lytic phage biology that have heretofore been intractable with in vivo genetic engineering methods. This manuscript describes that the presence of λ Red recombination proteins makes in vivo recombineering of T7 possible, so that single base changes and whole gene replacements on the T7 genome can be made. Red recombination functions also increase the efficiency of T7 genome DNA transfection of cells by ~100-fold. Likewise, Red function enables two other T7-like bacteriophages that do not normally propagate in *E. coli* to be recovered following genome transfection. These results constitute major technical advances in the speed and efficiency of bacteriophage T7 engineering and will aid in the rapid development of new phage variants for a variety of applications.

## 1. Introduction

Recently, interest has grown in rapid, efficient, and scalable engineering of bacteriophages, particularly virulent phages like T7 [1]. Increased interest in bacteriophage therapy has focused attention on T7 [2] due to its rapid growth, large burst size (~250 particles per infection), general resistance to restriction by the bacterial host, and rapid degradation of the host genome [3]. Phage T7 has been studied intensively for more than fifty years. The viral genome sequence and most gene functions have been defined. When T7 infects E. coli strains, rapid cell death and lysis occurs, usually within twenty minutes, with the release of progeny phage particles that spread the infection. Under optimized conditions, the time period from phage infection to host cell lysis is less than 20 min at 37 °C [4,5]. T7 is strictly lytic and is not capable of transduction, except under very special conditions [6]. Lytic or virulent phages are excellent models for developing phage therapy systems for use in combating antibiotic resistant bacteria. However, many of the attributes that make T7 a preferred system for phage therapy applications also present challenges. Standardized methods for in vivo genetic engineering, like that of recombineering, have not yet been successful using the T7 genome as a target. Here we introduce novel methods for recombineering the T7 genome and detail several intrinsic genes that are useful in the selection of recombinants.

One such gene, The bacterial *trxA* gene encodes thioredoxin, which is required by the T7 DNA polymerase as a processivity factor. An *E. coli* host with a defect in *trxA* blocks wild-type T7 phage growth, however, when *trxA* is cloned into T7, its replication and development are restored. This property allows *trxA* to be used as a genetic marker for positive selection of *trxA* recombinants into T7 genomes. This genetic approach is analogous to using antibiotic resistance genes for selecting recombination events in a bacterial genome [7]. However, even with a selectable marker, engineering virulent bacteriophages like T7 can be complicated, expensive, and time consuming. Past T7 engineering strategies have included standard in vitro molecular cloning and in vitro packaging of the genome [1], cloning and transformation of a plasmid carrying desired markers followed by infection of T7 phage to allow homologous recombination between the plasmid and the phage [8], and in vitro synthesis and assembly of full length genomes carried on yeast artificial chromosomes [8]. Here, we used the Red recombination system from phage λ as a tool to rapidly engineer genetic elements efficiently and with high-fidelity into T7 in vivo. It has already been shown that certain bacteriophages can readily be modified using BRED technology or other recombineering systems [9,10]. The λ Red recombineering system consists of a 5′-3′ double-strand DNA (dsDNA) exonuclease (Exo) and a single strand binding protein that anneals complementary DNA strands (Beta) [11]. A third function, Gam, is also commonly expressed in conjunction with λ Exo and Beta, and although not essential for recombineering, it stimulates dsDNA recombination about 10-fold [12] by disabling the potent bacterial RecBCD exonuclease and thereby helping to preserve linear dsDNA substrates electroporated into the cell [13]. Here, we have optimized the Red system specifically for modification of the bacteriophage T7 genome.

T7 genomic DNA can be introduced into cells by either infection or electroporation [14]. During phage T7 infection, the phage DNA initially enters the cell by injection of the left end of the genomic DNA, which is subsequently drawn further into the cell by coupling DNA entry to transcription by both the host and phage RNA polymerases [15]. This process establishes a temporal order of gene entry and expression from the left-most early genes to the distal late genes. The timing of DNA replication is also dependent on this mechanism of entry [16,17,18]. By contrast, when the entire genome is transferred into the cell by electroporation, the relative timing of both gene expression and DNA replication is altered. Here we find that T7 DNA, introduced by either infection or electroporation, can be genetically modified in vivo by recombineering using λ Red-mediated recombination. We also demonstrate that the efficiency of phage recovery following electroporation of T7 and other T7-like phage genomes is enhanced by λ Red recombination functions. The methods presented here provide new avenues for making synthetic phage variants that can be employed for different applications.

## 2. Results and Discussion

### 2.1. λ Red-Mediated Recombination Enhances Transfection of T7 and T7-Like Phages

T7 infection of *E. coli* is a multi-step process. After the phage attaches to the cell surface, its DNA is injected by a specialized event that requires transcription-mediated mobilization of the DNA from the viral particle into the cell by the host RNA polymerase [15]. In our initial experiments, we tested if introduction of purified phage genomic DNA by electroporation could produce functional phage particles. This process, also referred to as transfection, yielded very few plaques. The plaque yield could be increased several-fold by deletion of the *E. coli* K12-specific restriction endonuclease gene, *hsdR*. More importantly, we found that expression of the λ Red recombination genes increased T7 transfection efficiency by greater than 10 orders of magnitude (Figure 1A). Perhaps due to faster recombination and better preservation of the page genome.

Next, we found that the λ Red-mediated enhancement of T7 phage yields following genome electroporation extends to other T7-like bacteriophages. Bacteriophage K1-5 specifically infects *E. coli* K1 and *E. coli* K5 strains. Phage K1-5 cannot attach to and productively infect *E. coli* K12 [19]. In this experiment, plaque formation on *E. coli* K1, the natural host of the K1-5 phage, is used to determine if viable phages are generated following electroporation of the K1-5 genome into *E. coli* K12. No plaque-forming K1-5 phages were recovered after K1-5 genomic transfection of *E. coli* K12 strains. However, reasonable yields (~10^3^) of plaque forming derivatives were found after transfection of *E. coli* K12 strains expressing the Red system (Figure 1A).

Transfections by two other T7-like phage genomes, Pharr and Poteet, whose cognate host is Klebsiella pneumoniae were also tested. Red-mediated transfection of DNA genomes isolated from Pharr and Poteet yielded 10^3^ to 10^4^ plaques in Klebsiella when λ Red function was provided, compared to <10 plaques in the absence of Red (Figure 1A).

Quantitative results demonstrated that phage yields during transfection of T7 genomic DNA increase coordinately with increasing genomic DNA transfected per culture and saturate when there is approximately one T7 genome transfected per λ Red expressing cell (Figure 2). These data suggest that the Red system has its stimulatory effect on a single T7 genome and does not require two genomes to enter the same cell. If the latter were true, we would expect a strong stimulation of yield when the number of T7 genomes transfected approached the number of cells being transfected. Instead, there appears to be a slight decrease in yield when the number of genomes transfected increases.

### 2.2. T7 Transfection Requires the Exo and Bet Gene Functions and Is Independent of the Host RecA Function

We examined phage yields following T7 DNA transfection of *recA* minus strains that either expressed all the λ Red functions *exo*, *bet* and *gam* or were defective for *exo* and *bet*, but expressed gam. When Exo and Beta were present, high levels of transfection occurred even in the *recA* mutant. In the absence of Exo and Beta, transfection was greatly reduced (Figure 1B).

λ Red functions are known to be able to generate circular DNAs from linear DNA molecules having terminal homologous repeats [20]. T7 phage carries 160 bp DNA repeats at the ends of its genome and could thereby be circularized by Red recombination. Exactly how such a recombinant T7 circular DNA leads to T7 gene expression and replication remains to be determined, but it is known that the early T7 promoter is effectively transcribed by the host RNA polymerase on circular DNA [21]. This transcription leads to early gene expression and the potential for lytic development.

### 2.3. λ Red-Mediated Oligo Recombination Enables Directed Nucleotide Changes within the Phage T7 Genome

We next tested the ability of Red-mediated recombination to modify the T7 genome at precise locations. We first focused on *g17*, which encodes the tail fiber protein, a primary factor in determining the host range of T7 [7]. Starting with a T7 phage that contained a nonsense (amber) mutation in the tail fiber gene, *g17_am267_* [5], we tested how efficiently it could be converted to wild-type *g17* by a 71nt single-strand DNA (ssDNA) oligo containing the wild type sequence. We used one or the other of two complementary oligos: one (ARP292) corresponding in sequence to the leading strand, and its complement (ARP293) corresponding to the lagging strand used during T7 DNA replication (see Table 2). Initially, we focused on recombineering using transfected T7 genomic DNA. Strain ARP3771s carries an amber tRNA suppressor allele plus the λ Red recombination genes. This strain was made competent for recombineering and then electroporated with the mutant genomic T7 *g17_am267_* DNA (50 ng) together with one of the amber mutation-correcting 70 nt oligos. The electroporated cells were allowed to recover and the resulting lysate was titered on *E. coli* B and *E. coli* B40. Since *E. coli* B40 contains the tRNA suppressor allele, *supD*, all phage produced by DNA transfection should plate on this strain, regardless of the *g17* allele. Only the phages that are recombinant for the *g17* amber mutant and are now wild type for the tail fiber gene *g17* should grow on *E. coli* B. We calculated the frequency of these wild type recombinants over a range of oligo concentrations, keeping the T7 genomic DNA concentration high and constant (Figure 3). To monitor the spontaneous reversion of the amber mutation to wild type, we included a control in which only the T7 genomic DNA was added (see legend to Figure 3). No revertants were found among total transfectants indicating a reversion frequency of less than 2.3 × 10^−5^.

Both oligos generated wild type recombinants with the lagging strand oligo having a higher number of recombinants than the leading strand oligo (Figure 3). The sequence of the lagging strand oligo corresponds to that of Okazaki fragments made during the process of T7 DNA replication of gene *g17* [18]. This finding is consistent with other reports that demonstrate Red-dependent oligo recombination occurs primarily at the replication fork, and recombination frequency on the lagging strand is higher than on the leading strand [11].

A dsDNA PCR product amplified from the wild-type T7 gene *g17* was also tested and shown to generate wild type T7 recombinants (Figure 3). The dsDNA substrate was more efficient than ssDNA oligonucleotides at the lower DNA concentrations. We expect this result since double-strandedness protects this DNA from degradation by the multiple single strand exonucleases in *E. coli*. We know that ssDNA oligos are very susceptible to these single strand exonucleases, and this is most apparent at low oligo concentrations. Sawitzke et al. [22] demonstrated that high oligo concentrations were necessary to titrate the single strand exonuclease activities and thereby protect oligos so that they are available for recombination.

We have found that λ recombineering can be used to modify the T7 genome following transfection of its DNA genome and also following infection by the T7 phage particle. Cells expressing λ Red functions were infected with the tail fiber mutant phage T7 *g17_am267_.* Lysates of this phage were prepared in the host *E. coli* B40, which contains a tRNA suppressor for the amber mutation. The suppressor allows the amber mutation to be translated and thereby produce the essential GP17 protein.

Cells being prepared for recombineering were mixed with the T7 *g17_am267_* mutant phage to allow adsorption during the time that they were on ice prior to recombineering. Unabsorbed phages were eliminated by washing steps. The electroporation conditions and the substrate oligos used for recombineering were the same as those used for the transfection and recombineering of phage genomic DNA. The products of the recombineering reactions were assayed on both *E. coli* B and B40 to determine the frequency of wild type recombinants (see Methods, Figure 4). Spontaneous reversion of the amber mutant to wild-type was observed at a frequency of approximately one in 10^6^ viable phage (see Figure 4 and its legend), which is consistent with previous reports documenting the reversion frequency of T7 amber mutants [5,23,24]. Providing the oligo without λ Red functions present revealed that phage T7′s own recombination and replication functions have little if any ability to mediate oligo recombination, whereas λ Red functions greatly enhance targeting T7 genome recombination by as much as 10,000-fold at high oligo concentration (Figure 4).

### 2.4. Gene Replacement by λ Red-Mediated Recombineering

Thioredoxin is the product of the bacterial *trxA* gene. TrxA is a processivity factor for the T7 DNA polymerase and is therefore essential for T7 [25]. Thus, wild-type T7 phage cannot form plaques on *E. coli* if the *trxA* gene is deleted from the bacterial chromosome. However, if the *trxA* gene is incorporated into and expressed from the T7 genome itself, phage replication and plaque forming ability can be restored even if the *E. coli* strain is deleted for *trxA*.

We have replaced the non-essential T7 nucleotide kinase gene *g1.7* [7] with the *trxA* gene by recombineering. In this experiment, the entire T7 genome was introduced into cells by electroporation along with a double-stranded PCR product of *trxA* to target replacement of T7 gene *g1.7*. Recombineering to insert and replace *g1.7* with the *trxA* DNA was very efficient (0.6%) (Figure 5).

We have shown above that a non-essential T7 gene (*g1.7*) can be replaced very efficiently with *trxA* using recombineering. We now wanted to replace the essential tail fiber gene, *g17*, with *trxA* and used recombineering to generate recombinant T7 *g17*<>*trxA* phages. Because *g17* makes the tail fibers used for phage attachment to the cell surface, T7 recombinants lacking *g17* will not be able to attach and infect *E. coli*. Therefore, to detect these recombinants, we constructed a special strain of *E. coli*, ARP3750, which expresses Gp17 tail fiber protein constitutively (see methods). Wild-type T7 cannot grow on this *trxA* mutant strain since the bacterium and wild type T7 lack *trxA*. A T7*g17*<>*trxA* recombinant phage carrying *trxA* would be able to grow because the essential Gp17 tail fibers are expressed from the bacterial chromosome. Figure 5 shows that in this strain, *trxA* recombinants replace the essential gene *g17* at a frequency of ~1.0%.

We have observed a ~1.0% frequency of gene replacement by *trxA* at the essential *g17* tail fiber locus, and a similar 0.6% frequency of replacement by *trxA* at the nonessential *g1.7* locus (Figure 5). The generation of tail fiber recombinants in which the *g17* tail fiber gene of phage T7 is replaced by other bacteriophage type tail fiber genes can now be detected by screening a few thousand plaques generated after transfection. Recombineering methods for exchanging tail fiber genes opens the potential for extending T7′s host range and thereby weaponizing T7-type phages against drug resistant bacteria [26]. Indeed, Yehl et al. [27] recently demonstrated that they could extend the host range of bacteriophage T3 by mutagenizing the tail fiber gene. These works are complementary, as engineering the mutants would be greatly facilitated by using recombineering methods described herein.

Lytic or virulent phages are excellent models for developing anti-bacterial systems to combat antibiotic resistant bacteria. Here we have engineered the T7 genome with the goal of generating a general platform for modification of T7 that can be easily manipulated and has potential for biomedical applications.

## 3. Materials and Methods

### 3.1. Media

For standard cultivation of bacteria, cells were grown in liquid L Broth (LB) containing 0.5% NaCl, 1.0% (*w/v*) tryptone, and 0.5% (*w/v*) yeast extract. LB Agar plates contained in addition 1.5% agar. For sucrose counter-selection, NaCl was omitted from LB agar and 6% (*wt/vol*) sucrose was added. To select for antibiotic resistant genetic markers, the following antibiotic concentrations were used: 12.5 μg/mL tetracycline, 30 μg/mL ampicillin, and 200 μg/mL hygromycin.

SOC medium was used for recovery of cells following electroporation, and contained 2% (*w/v*) tryptone, 0.5% (*w/v*) yeast extract, 10 mM NaCl, 2.5 mM KCl, 10 mM MgCl_2_, 20 mM glucose. Phages were diluted in TMG, containing 10 mM Tris-HCl, pH 7.5, 10 mM MgSO_4_, and 0.1% gelatin. Plaque assays were carried out on Tryptone Broth (TB) agar plates containing 0.5% NaCl, 1.0% (*w/v*) tryptone, and 1.0% (*w/v*) agar.

### 3.2. Bacterial Strains

Bacterial strains used in this study are listed in Table 1. Strains specific to this paper were generated by recombineering and/or P1 transduction. The method for strain construction is provided in Table 1. All oligos used in strain construction were purchased from Integrated DNA Technologies (IDT, Coralville, IA, USA) and are listed in Table 2. Final constructions were verified by PCR analyses and sequencing. Details of individual strain constructions will be provided upon request.

The deletion *trxA<>tetA* was constructed on the bacterial chromosome of LT976 by recombineering to replace *trxA* with *tetA* [28] to generate ARP3620. The *tetA* cassette was amplified from T-SACK [29] using primers ARP309 and ARP310, which add 50 bp of homology from the regions flanking the *trxA* gene to the flanks of the *tetA* cassette. The *tetA* insertion in ARP3620 was moved by P1 transduction [30] to generate the *trxA* deleted strains *E. coli* B3783 and ARP3750.

To cultivate T7 phages that lack *g17*, which encodes the essential tail fiber protein, we constructed strain ARP3607 to express *g17* in trans. The *g17* coding region, flanked by λ homologies, was generated by PCR using oligos ARP261 and ARP262. This linear fragment was inserted into a defective λ prophage by recombineering in strain LT976 to allow expression of Gp17 from the *pL* promoter of λ [12].

### 3.3. Bacteriophages

Bacteriophages used in this study are listed in Table 1. T7 phage was obtained from the collection of Ian Molineux, and Klebsiella phages Pharr and Poteet were isolated in collaboration with the laboratory of Jason Gill and Ry Young at Texas A&M. Phage K1-5 was a laboratory stock of S. Adhya. Pharr and Poteet genomic DNAs were a generous gift from Jason Gill.

All phage lysates were grown in TB by diluting an appropriate overnight bacterial culture 1:5 into fresh TB and adding a single phage plaque picked up by Pasteur pipet. Cultures were grown until lysis occurred, then chloroform was added, and cell debris was cleared by centrifugation.

### 3.4. Preparation of Genomic DNA

Genomic DNAs of ~40 kb in length from the phages T7, Pharr, Poteet, or K1-5 were extracted from lysates, having titers of ~1.0 × 10^10^ PFU/mL, using the Qiagen lambda Midi Kit (Qiagen, Valencia, CA, USA) according to the manufacturer’s recommendations. Phage genomic DNA was diluted to a working concentration of 100 ng/µL in ddH2O for all subsequent manipulations.

### 3.5. Recombineering Using Transfected Bacteriophage Genomic DNA

Red-recombination proficient cells (ARP3771) were made competent for Red recombination and electroporation as described in [28]. Briefly, log phase cells were shifted from a 32 °C to a 42 °C shaking water bath for 15 min to induce λ Red functions and subsequently cooled in an ice bath to prepare the cells for electroporation. Electroporation reactions were prepared [31,32] by mixing 50 μL of cells with 50 ng genomic DNA); when required, ssDNA or dsDNA substrates containing the desired genetic changes were added. The cells and DNA were transferred to a pre-chilled 0.1 cm electroporation cuvette (Bio Rad, Hercules, CA, USA) and electroporated at 1.3 kV in a Bio Rad Micropulser (Bio Rad, Hercules, CA, USA); the voltage used for electroporation was decreased to 1.3 kV, (below the standard 1.8 kV used for *E. coli*) to accommodate the large size of T7 genomic DNA [33]. The electroporation mix was allowed to recover in 10 mL SOC medium and processed for either infective centers or total phage yield as described below.

To assess the number of infectious centers following transfection, cells were diluted from the electroporation into 10 mL SOC and shaken for 15 min in baffled flasks in a 32 °C water bath, before pelleting at 15,000× *g* for 1 min. Cell pellets were suspended in 1 mL TB and then serially diluted in TMG. These dilutions were mixed with plating bacteria and layered on TB agar plates to determine the number of transfected cells that generate an infectious phage resulting in a plaque.

When monitoring phage yield after complete lysis, electroporation reactions were mixed with 10 mL SOC and incubated in a 32 °C shaking water bath for 60 min, or until lysis, and at which time lysis was completed by addition of chloroform before centrifugation to remove cell debris. Resulting lysates were diluted and assayed on selective and non-selective hosts (see below), respectively, to calculate recombination frequencies, relative to total phage yield.

### 3.6. Recombineering Following Infection by Bacteriophage T7

Red-recombination proficient cells (ARP3771) were prepared as described above to express the Red proteins from the defective λ prophage, and 100 μL of a T7 phage lysate containing 2.4 × 10^10^ PFU/mL were added to the 15-mL recombination competent cultures, equivalent to an average multiplicity of infection of 5. After 15 min, cells were washed as described above to prepare them for electroporation and to remove free phage. Electroporation reactions were prepared by mixing 50 μL of electrocompetent cells, 0.5 μL of single-strand or double-strand DNA containing the desired genetic modification. The cells and DNA were transferred to a pre-chilled 0.1 cm electroporation cuvette and electroporated at 1.8 kV. Cells recovered in 10 mL SOC medium by incubation at 32 °C for 1 h at which time each culture received chloroform and was centrifuged to remove cell debris. The resulting cleared lysates were assayed on non-selective and selective strains of *E. coli* to determine total and recombinant phage titers (details below).

### 3.7. Quantitation of Phage in Lysates

Phage titers were determined by plaque assays and were carried out by infecting 200 μL of host bacteria grown overnight in LB with 100 μL of phage lysates diluted appropriately in TMG buffer. After 15 min for adsorption, the infected bacteria were mixed with 3 mL of pre-warmed (50 °C) tryptone broth agar (0.7% agar) and layered over TB-agar (1% agar) plates and incubated at room temperature overnight. *E. coli* and its derivatives were used as host for T7 phage; *E. coli* K1 was used as host for phage K1-5 phage; and *Klebsiella pneumoniae* strain 1760b was used as host for phages Pharr and Poteet. T7 recombinants containing an insertion of the *trxA* gene were titered on *E. coli* B3783, and T7 containing amber mutations were titered on *E. coli* B40.

## 4. Conclusions

T7 and other T7-like bacteriophages are valuable for their innate virulence and bacteria-killing properties. T7 is used as a model for lytic phage biology, and its genetics have been studied for over 50 years. Historically, however, virulent phages have been difficult to modify by in vivo genetic engineering methods. We have developed λ Red recombineering for T7 and used the Red functions to engineer changes in essential and non-essential genes of the phage. Additionally, we found that the Red recombination functions stimulate T7 phage production following transfection of T7 genomes into *E. coli*. This latter feature is important since the stimulation applies not only to phage T7 that uses *E. coli* as its natural host, but also to T7-like phages Pharr and Poteete that normally infect the pathogenic host Klebsiella pneumoniae where Red-mediated recombineering is inefficient [34]. By enabling their transfection and development in *E. coli*, research has been extended to these phages in a non-pathogenic environment. The methods presented here expand the tools available for engineering T7-like bacteriophages and enable facile construction of novel bacteriophages for basic research purposes and new therapeutic applications.

## Figures and Tables

**Figure 1 antibiotics-09-00805-f001:**
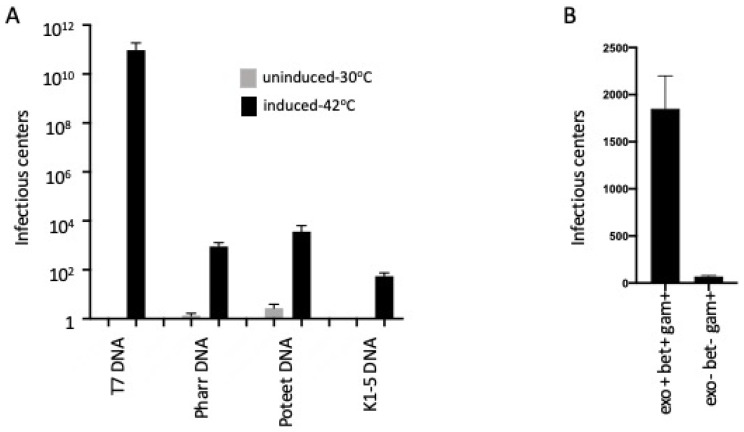
λ Red function enhances transfection by T7 and T7-related phage genomic DNA. (**A**) LT976 strain contains a λ defective prophage that expresses Red recombination functions only when induced at 42 °C (see Table 1). The light-colored bars indicate cells grown at 32 °C without Red expression; the dark-colored bars indicate cells induced at 42 °C to express Red recombination functions. Standard error of the mean value is shown and indicates *p* < 0.005 in Student *t*-test, comparing induced samples to uninduced controls. Cells, either induced at 42 °C or maintained at 32 °C, were electroporated with ~50 ng of respective phage DNAs (see X-axis, ~1 phage genome per cell). The phage DNA was introduced into LT976 by electroporation as described in Materials and methods. The Y-axis indicates total number of infectious centers per electroporation reaction. The phages Pharr and Poteet were scored on their cognate host Klebsiella pneumoniae 1760b, and the phage K1-5 was scored on its cognate host *E. coli* K1 (see Table 1). Note that without induction of Red (32 °C no transfectants were found for phage K1-5. (**B**) In a separate experiment, the RecA defective strain DH5α carries the plasmid pSIM18 or pSIM28. These plasmids contain a defective prophage with genes to express Exo, Beta and Gam or just Gam, respectively. Expression is induced by a shift to 42°. Transfection of the T7 genome is as described in Methods. Bars show level of phage yields following transfection. Negligible infective centers are found in the absence of Exo and Beta.

**Figure 2 antibiotics-09-00805-f002:**
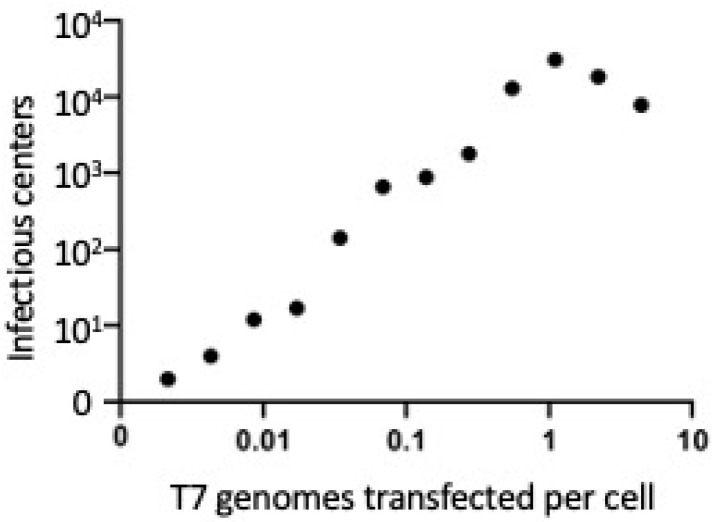
The number of infectious centers recovered from electroporated T7 genomic DNA increases with concentration. One nanogram of T7 genomic DNA is approximately equivalent to 2.3 × 10^7^ genomes, and each electroporation reaction contains ~1 × 10^8^
*E. coli* host cells (LT976). Thus 50 ng of phage DNA = ~1.1 genomes/cell. Electroporation was carried out on cells induced for λ Red recombination functions, then incubated in LB for growth at 30°, before plating on TB agar in the presence of *E. coli* B plating bacteria for determining total plaque forming units at each genome concentration used. Transfection reaches saturation where one T7 genome is transfected per cell. Infectious centers generated per electroporation reaction are indicated on the Y-axis. Addition of >1 genome/cell appears to be somewhat inhibitory to recombineering and may represent competition of a second molecule for circularization.

**Figure 3 antibiotics-09-00805-f003:**
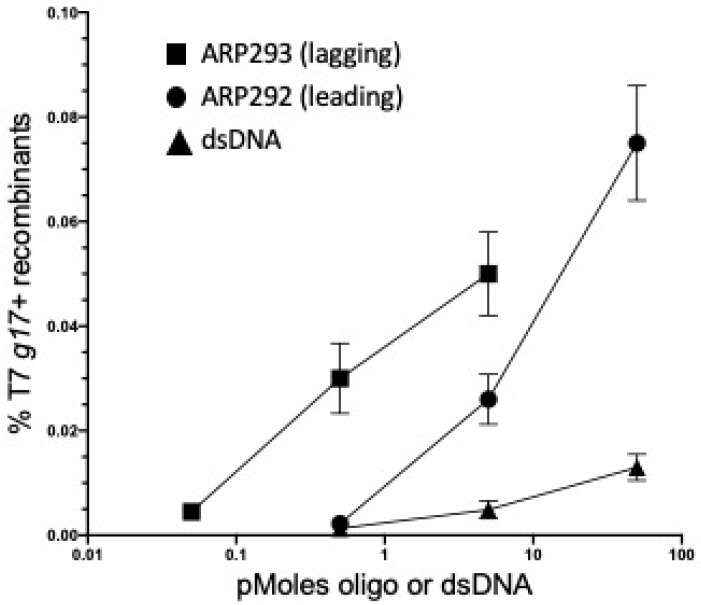
Recombineering can correct a point mutation following transfection of the T7 genome. The T7 *g17_am267_* mutant genomic DNA (50 ng) was co-electroporated with the substrate DNAs (pmole amounts on X-axis) into the amber suppressor strain ARP3771 expressing the λ Red functions. Single-strand oligonucleotide substrates ARP292 (circles) and ARP293 (rectangles) of 71 nucleotides in length carry the wild type coding information. Their sequences correspond, respectively, to the leading and lagging strands used in T7 DNA replication. The dsDNA substrate (triangles) is a PCR product corresponding in length and sequence to the single-stranded oligos used here. T7 *g17^+^* wild type recombinant frequencies (Y-axis) were calculated by the following equation: (phage titer on *E. coli* B)/(phage titer on *E. coli* B *supD*). Values represent an average of at least three biological replicates with standard errors of the mean indicated. A control T7 genome transfection without added DNA substrate generated no wild type T7 revertants among an average of 4.4 × 10^4^ total plaques recovered on *E. coli* B *supD* in eight experiments, indicating a reversion frequency < 2.3 × 10^−5^.

**Figure 4 antibiotics-09-00805-f004:**
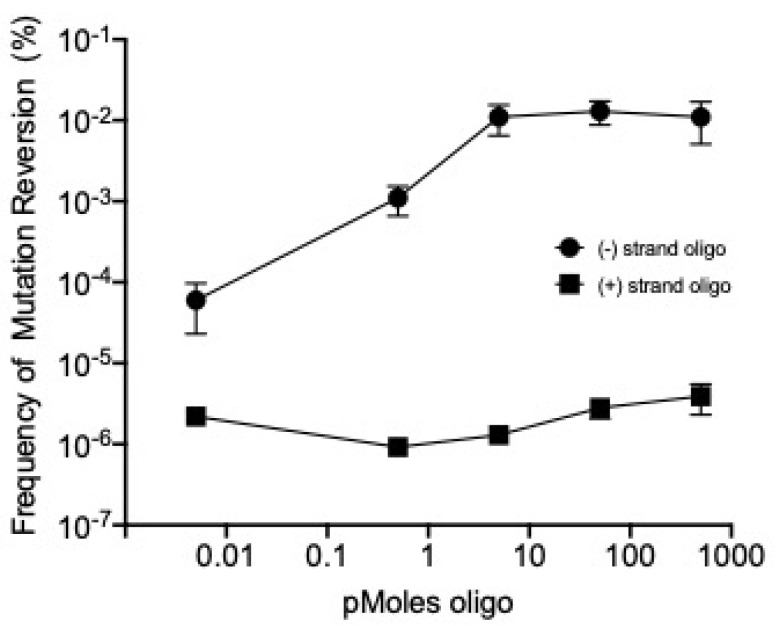
Recombineering corrects a point mutation after infection by a T7 phage particle. The lagging strand single-strand oligonucleotide (ARP293) of 71 nucleotides in length was used as a substrate during recombineering to correct the phage mutation T7 *g17_am267_*. The amber suppressor strain ARP3771 was infected with the T7 *g17_am267_* mutant phage at a multiplicity of infection of ~5 phage per bacterium. Cells were either induced to express Red function and then electroporated with the leading and lagging strand oligonucleotides ARP292 and ARP293 at several oligo concentrations (see *x*-axis). Electroporation reactions were outgrown at 37 °C in SOC media until lysis. Frequency of *T7 g17^+^* recombinants (y-axis) was calculated by the following equation: (recombinant phage titer on *E. coli B*)/(total phage titer on *E. coli* B40 *supD*). Values represent an average of at least three biological replicates. Lysates, in which no lagging strand oligo DNA was added, revealed a reversion frequency of ~1.1 × 10^−6^ (±4.1 × 10^−7^).

**Figure 5 antibiotics-09-00805-f005:**
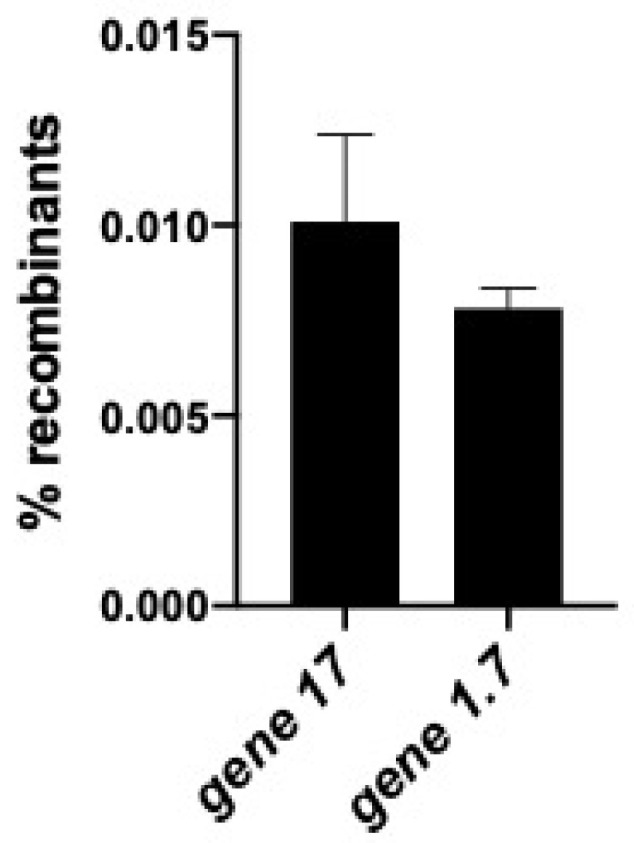
Replacement of T7 genes *g1.7* and *g17* with the bacterial *trxA* gene. Recombineering was used to replace two different T7 genes with the bacterial *trxA* gene. One gene, *g1.7*, is non-essential; the other gene, *g17*, is essential and required for plaque formation. The *trxA* dsDNA PCR fragments, *g1.7*<>*trxA* and *g17*<>*trxA* were amplified using the bacterial t*rxA* gene as a template to generate the *trxA* gene flanked by T7 DNA homology that targets recombination to *g1.7* or *g17*, respectively. Oligos ARP301 and ARP302 used for PCR are designed with *trxA* having T7 flanking homologies to replace *g1.7* with *trxA*. Oligos ARP356 and ARP357 used for PCR are designed with *trxA* having T7 flanking homologies to replace *g17* with *trxA* (Table 2). LT976 cells were induced for Red-mediated recombination and then co-electroporated with genomic T7 DNA and either the *g1.7* or *g17* targeted *trxA* PCR products. The electroporation reactions were out-grown in SOC medium at 37 °C. Gene 1.7 *trxA* replacements were detected on *E. coli* B 3783, a host that lacks the *trxA* gene in its genome. The *g17* replacements were detected on ARP3750 at 42 °C. Non-recombinant T7 cannot grow on these two bacteria that lack *trxA*. T7 recombinant frequencies in the lysates were calculated by the following equations: [(T7 *g1.7*<>*trxA* recombinant phage titer on *E. coli* B3783 Δ*trxA*/(total phage titer on *E. coli* B)] × 100; [(T7 *g17*<>*trxA* recombinant phage titer on ARP3750)/(total phage titer on ARP3607)] × 100. Values represent an average of at least three biological replicates with the standard error of the mean calculated. The identity and accuracy of the gene substitutions on the recombinant phages were confirmed by sequence.

**Table 1 antibiotics-09-00805-t001:** Bacterial strains, Bacteriophages, and Plasmids used in this study.

Bacterial Strains	Relevant Genotype	Source, Reference
*E. coli* strains		
*E. coli* B	Wild-type	Ian Molineux collection
*E. coli* B40	*E. coli* B *supD*	Ian Molineux collection
*E. coli* B3783	*E. coli* B Δ*trxA*<>**tetA*	This work; recombineering pSIM18; gene replacement *hsdR*<>*bla*
*E. coli* K1	Wild-type	Sankar Adhya collection
*E. coli* K12 strains:		
W3110	*rph*-1 *inv*(*rrnD*-*rrnE*)	Laboratory stock strain
DY330	W3110 [λ cI857 *gam*^+^*bet*^+^*exo*^+^ Δ(*cro*-*bioA*)]	{Thomason et al., 2014}
T-SACK	W3110 *araD*<>*tetA*-*sacB*-amp *fliC*<>*cat argG*:*kan*	{Li et al., 2013}
LT976	W3110 *hsdR*<>*bla* λ cI857 *gam*^+^*bet*^+^*exo*^+^ Δ(*cro*-*bioA*)	Lynn Thomason collection; recombineering; DY330, gene replacement *hsdR*<>*bla*
ARP3607	W3110 *hsdR*<>*bla* λ cI857 *pL g17* Δ(*cro*-*bioA*)	This work; recombineering; gene replacement LT976, (*cIII*-*galK*)<>*gp17*
ARP3750	W3110 *hsdR*<>*bla* Δ*trxA*<>*tetA* λcI857*pL g17* Δ(*cro*-*bioA*)	This work
ARP3620	LT976 Δ*trxA*<>*tetA*	This work
ARP3771	LT976 *supF*:Tn10	This work; transduction LT976 X (P1)
ARP3816	W3110 *hsdR*<>*bla*	This work; transduction W3110 X (P1) LT976
DH5α	*fhuA2 lac(*del*)U169 phoA glnV44* Φ80’ *lacZ(*del*)*M15 *gyrA96 recA1 relA1 endA1 thi-1 hsdR17*	Laboratory stock strain
*Klebsiella pneumoniae* Kpn1760	Cured of resistance plasmid pKpQIL at 42 °C	Karen Frank collection
Bacteriophages and Plasmids		
T7	Wild-type	Sankar Adhya collection
T7 (amber267)	T7 *g17*_Q170UAG_	Ian Molineux collection
K1-5	Wild-type	Sankar Adhya collection
Pharr	Isolated from sewage	Jason Gill collection
Poteet	Isolated from sewage	Jason Gill collection
pSIM18 *HygroR*	pSC101 *P_L_gam^+^bet^+^exo^+^*	{Thomason et al., 2014}
pSIM28 *HygroR*	pSC101, *P_L_-gam^+^*	(http://redrecombineering.ncifcrf.gov/)

* “<>” represents a convention used by some bacterial geneticists to denote “[gene] replaced by [gene]” (e.g., *ΔtrxA*<>*tetA).*

**Table 2 antibiotics-09-00805-t002:** Oligonucleotides ^a^.

Name	Nucleotide Sequence ^b^	Notes
ARP261	TAACGCTTCACTCGAGGCGTTTTTCGTTATGTATAAAAAGGAGCACACC/ATGGCTAACGTAATTAAAACCGTTTTGACTTACCAG	λ *pL g17* forward oligo
ARP262	GCTTCCCAGCCAGCGTTGCGAGTGCAGTACTCATTCGTTTTATACCTCTG/ATTACTCGTTCTCCACCATGATTGCATTAGG	λ *pL g17* reverse oligo
ARP292	ACCAGAACTCATGGCAAGCACGTAATGAAGCCTTA CAGTTCCGTAATGAGGCTGAGACTTTCAGAAACCAA	T7 *g17*_Q170UAG-CAG_ leading strand oligo
ARP293	TTGGTTTCTGAAAGTCTCAGCCTCATTACGGAA CTGTAAGGCTTCATTACGTGCTTGCCATGAGTTCTGGT	T7 *g17*_Q170UAG-CAG_ lagging strand oligo
ARP301	GGCAGTGACCCGCTTCCCGTTCGTCCGTCTGTTACTCAAACGAATCAAGGAGGTGTTCTG/ATGAGCGATAAAATTATTCACCTGACTGAC	T7 *g1.7*<>*trxA* forward
ARP302	CACTCTGAGCAAGATGTGAAGTCATCAGATAGGCTGTCGGCAGGTGGGGTTGACTTGAAG/TTACGCCAGGTTAGCGTCGAGGA	T7 *g1.7*<>*trxA* reverse
ARP309	ACCAACACGCCAGGCTTATTCCTGTGGAGTTATAT/TCCTAATTTTTGTTGACACTCTATC	*trxA*<>*tetA* forward
ARP310	TTTTTAGCGACGGGGCACCCGAACATGAAATTCCC/ATCAAAGGGAAAACTGTCCATATGC	*trxA*<>*tetA* reverse
ARP321	ACCAGAACTCATGGCAAGCACGTAATGAAGC	*g17_am267_* 70 bp dsDNA forward
ARP322	CGCTTGGTTTCTGAAAGTCTCAGCCTCATTACGG	*g17_am267_* 70 bp dsDNA forward
ARP356	CGAAATAATCTTCTCCCTGTAGTCTCTTAGATTTACTTTAAGGAGGTCAA/ATGAGCGATAAAATTATTCACCTGACTGAC	*g17*<>*trxA* forward
ARP357	AGGTACAGTCATTGTTGTTATCTGACCCTCTACCAATGTACCAGTTATTC/TTACGCCAGGTTAGCGTCGAGGA	*g17*<>*trxA* reverse

^a^ Oligos were obtained desalted from Integrated DNA Technologies in Coralville, IA. ^b^ “/” indicates the border between 5′ homology to recombineering targets and 3′ homology used for primer PCR. The underlined CAG or CTG are the codon positions in the oligos ARP292 and ARP293 that correct the amber mutation in g17 used in these studies.

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
