# Peer review of "λ Recombineering Used to Engineer the Genome of Phage T7"

_antibiotics, 2020, doi:10.3390/antibiotics9110805_

Round 1
Reviewer 1 Report
This is a terrific report on engineering of the phage T7 genome. I have no major concerns or comments, and recommend publication, perhaps with some minor modifications.
Minor comments:
- The impact of the lambda Red system on T7 transfection is quite remarkable (10 orders of magnitude). It is speculated that this effect is mediated by genome recircularization, which seems reasonable, although less compelling that it is mediated by altered expression. Many other phages are able to transfect host cells without a Red-like requirement, even though they likely require circularization and have shorter repeats than T7. No revisions are really needed, but this is certainly a point of interest.
- The ability to efficiently engineer T7 is clearly important. The configuration where phage DNA and an oligonucleotide are co-electroporated (section 2.3) is reminiscent of the BRED strategy reported for mycophages (ref. 9), but there are seeming differences in that the proportion of recombinants seems much lower (Fig. 3) and at a frequency of less than 0.1% (if I interpret Fig. 3 correctly), selection would still be required to identify desired recombinants, and simple PCR screening would not be productive. In addition, the dsDNA substrate seems to work very inefficiently in this configuration. A comment on these differences might be useful.
- Similarly, it is stated that the efficiency of constructing the trxA recombinants is very efficient (lines 240, 270). However, the numbers reported in the text are 0.6% and 1.0%. Whether this is ‘very efficient’ may just be in the eye of the beholder! It is presumably much higher than without recombineering, but not really efficient enough to facilitate recombinant identification in the absence of selection, certainly not without extensive screening. The % recombinants in Fig.5 are reported as ~0.01%, and it is unclear how this relates to the 0.6% and 1.0% in the text. If they are calculated differently somehow, it would be helpful to standardize them.
- It is commented that virulent phages have been difficult to modify by in vivo genetic engineering methods (lines 370). Perhaps in the broadest sense this is true, but the BRED strategy mentioned above has been used not only in mycophages, but also E. coli, Salmonella, Klebsiella and Pseudomonas phages, and CRISPR-mediated methods have been reported recently too. Perhaps just adding some context from these more recent approaches might be helpful.
Author Response
Reviewer 1
1)We agree that lRed-mediated rapid circularization of the genome is a more likely explanation for the enhancement of phage recovery. This point is made on line 79.
2) The efficiencies measured in the mycophage BRED system and in our system reflect different measurements used to quantitate the products. The authors of the BRED system consider a positive signal as one where even a trace of the positive reaction is present in the plaque, whereas we are calculating the number of fully recombinant plaques in 108 cells. From our previous observations with the lambda red system, it is likely that the majority of all plaques would show traces of recombinants.
3) Additional commentary have been included to recognize that the BRED system also works on diverse bacteriophages.
Reviewer 2 Report
Major comments:
Ln 121: Why would you try to transfect more than one T7 genome into a cell?
Why was no efficiency of plating performed to test the engineered phages?
Was MOI measured for any of the experiments?
Why were incubation temperatures set at different values i.e. 30oC, 37oC, 42oC?
Author Response
Reviewer2
a)Ln121: The next to the last sentence of the previous paragraph describes our rationale for stating why we believe that no more than one phage is required. However, one can imagine a scenario where if more than one phage genome were necessary to produce a recombinant phage, we would have expected a stimulation of recombinants above one genome per cell, rather than the inhibition we see. One can imagine that two molecules could be required if the process of recombineering somehow damaged one genome.
b)EOP: permissive host transfection represents efficiency of plating. When correcting for infectious phage genomes, phage lysates were always plated on both permissive and non-permissive hosts to generate a frequency.
c)MOI: 50 ng of T7 phage DNA is the equivalent of 1.1 T7 genomes in 108 cells (moi=1.1)
d)The different temperatures were used to turn on the lambda red system (at 42oC), or keep it turned off (at 32oC). During outgrowth where lysis was not expected, the outgrowth was done at 37oC
Reviewer 3 Report
The manuscript of Jensen et al. presents the applicability of the red recombination system for the reconstruction of lytic T7 phage genomes. The content of the manuscript is highly interesting, however, there are some aspects that might be improved.
- The Latin name of bacterial has to be corrected throughout the manuscript into italic writing.
- The Figures: the legends are describing methodological things; the authors might consider preparing text shorter, easy to read and to put the other explanations in the methods.
- The gene names have to be written in italic as well.
- Some nonessential genes have been replaced with trxA. Please, add information on what are coding these non-essential genes and how did you decide to choose them.
- When describing the temperature, C is missing in some places. This must be corrected throughout the text.
- L302: The trxA deletion……
- L305: ……was removed
Author Response
Reviewer 3
- The Latin name of bacteria has been corrected throughout the manuscript into italic writing.
- The Figures: the legends are describing methodological things; the authors might consider preparing text shorter, easy to read and to put the other explanations in the methods. Some of the legends have been shortened.
- The gene names have been written in italic as well.
- Some nonessential genes have been replaced with trxA. Please, add information on what are coding these non-essential genes and how did you decide to choose them.
- When describing the temperature, C is missing in some places. This has been corrected throughout the text.
- L302: The trxA deletion……
- L305: ……was removed
6-7) This sentence has been modified to state: "The deletion trxA<>tetA was constructed on the bacterial chromosome of LT976 by recombineering (see Materials and Methods)."